# Unifying Generative and Dense Retrieval for Sequential Recommendation

## Abstract

Sequential dense retrieval models utilize advanced sequence learning techniques to compute item and user representations, which are then used to rank relevant items for a user through inner product computation between the user and all item representations. However, this approach requires storing a unique representation for each item, resulting in significant memory requirements as the number of items grow. In contrast, the recently proposed generative retrieval paradigm offers a promising alternative by directly predicting item indices using a generative model trained on semantic IDs that encapsulate items' semantic information. Despite its potential for large-scale applications, a comprehensive comparison between *generative retrieval* and *sequential dense retrieval* under fair conditions is still lacking, leaving open questions regarding performance, and computation trade-offs. To address this, we compare these two approaches under controlled conditions on academic benchmarks and propose LIGER (LeveragIng dense retrieval for GEnerative Retrieval), a hybrid model that combines the strengths of these two widely used methods. LIGER integrates sequential dense retrieval into generative retrieval, mitigating performance differences and enhancing cold-start item recommendation in the datasets evaluated. This hybrid approach provides insights into the trade-offs between these approaches and demonstrates improvements in efficiency and effectiveness for recommendation systems in small-scale benchmarks.

## 1 Introduction

Sequential recommendation methods (Kang & McAuley, 2018b; Zhou et al., 2020), have predominantly relied on advanced sequential modeling techniques (Hochreiter & Schmidhuber, 1997; Vaswani et al., 2017; Radford et al., 2019) to learn dense embeddings for each item and user. Using these embeddings, the most relevant items are retrieved through maximum inner product search. Despite its effectiveness, this approach requires comparing every item in the dataset during the retrieval stage, which can be computationally expensive as the number of items grows. Furthermore, each item must be represented by a unique embedding, which needs to be learned and stored, adding to the complexity.

In contrast, *generative retrieval* (Rajput et al., 2024) is a new approach, which deviates from the embedding-centric paradigm. Instead of generating embeddings, this approach utilizes a generative model to directly predict the item index. To better capture the sequential patterns within item interactions, items are indexed by "semantic IDs" (Lee et al., 2022a), which encapsulate their semantic characteristics. During the recommendation process, the model employs beam search decoding to predict the semantic ID (SID) of the next item based on the user's previous interactions. This method not only reduces the need for storing individual item embeddings but also enhances the ability to capture deeper semantic relationships within the data. Additionally, adjusting the temperature during generation can naturally produce more diverse recommendations.

To distinguish the first paradigm from *generative retrieval*, which also leverages sequential information, we refer to the embedding-centric paradigm in this paper as *(sequential) dense retrieval*. The term *dense retrieval* is borrowed from the information retrieval domain (Tran & Yates, 2022), with "sequential" added to differentiate it from other dual-encoder-based architectures Yi et al. (2019). The *generative retrieval* paradigm is well-positioned for future scaling in industrial recommendation

Figure 1: *Performance Comparison Between the Implemented Generative and Dense Retrieval Methods Across Datasets.* Dense retrieval computes the inner product between predicted item representations and the entire item set, scaling with $\mathcal{O}(N)$ and requiring storage for $\mathcal{O}(N)$ embeddings. In contrast, generative retrieval stores only $\mathcal{O}(t)$ learnable embeddings and predicts the next item using beam search, scaling with $\mathcal{O}(tK)$, where $K$ is the beam size and $t$ is the number of Semantic IDs. Using identical item content information, both methods were trained on various datasets, and their performance, measured by Recall@10, is reported in the table on the right. While the implemented generative retrieval method reduces computational and storage costs, it shows lower performance compared to the implemented dense retrieval method in the datasets we evaluated.

systems (Singh et al., 2023), offering significant savings in storage and inference time. However, while recent works continue to advance the dense retrieval paradigm (Hou et al., 2022c), generative retrieval methods are increasingly being integrated with pretrained models such as LLMs (Cao et al., 2024b) to improve item recommendation.

Despite these advancements, there is a notable lack of direct comparisons under equivalent conditions, raising questions about which paradigm performs better given the same input information and how their performance balances against storage and computation trade-offs. Given the focus on academic benchmarks and limited resources, we narrow our study to small-scale datasets commonly used in research (Rajput et al., 2024; Cao et al., 2024a; Liu et al., 2024b; Li et al., 2023b; Hou et al., 2022c; Zheng et al., 2024). Within this context, we aim to explore this question by comparing two representative implementations of sequential generative and dense retrieval models, ensuring consistency in input information and experimental setups.

To ensure a meaningful comparison, we made significant efforts to faithfully implement the TIGER method following the setup described in Rajput et al. (2024), conducting extensive hyperparameter tuning to optimize its performance. For the dense retrieval approach, we aligned its design with TIGER to ensure consistency in input information (adopting the transductive setting (Hou et al., 2022c)) and experimental setups. As shown in Figure 1, in our experiments, the dense retrieval approach demonstrates stronger performance than generative retrieval on both in-set and cold-start item predictions across the datasets we tested. Specifically, we observe that the generative retrieval method struggles with cold-start items, achieving close-to-zero performance on most datasets except for the Amazon Sports dataset. These observations are not entirely surprising, as dense retrieval methods have been well-established and extensively developed over time, excelling at learning high-quality embeddings that are particularly effective for recommendation tasks. In contrast, generative retrieval represents a relatively new paradigm that has yet to undergo the same level of refinement and optimization. Nonetheless, the generative retrieval approach holds significant potential (Singh et al., 2023), and future advancements could enable it to rival or even surpass dense retrieval under appropriate conditions.

At the current stage, to harness the strengths of both paradigms, we propose a novel hybrid model, LIGER, that combines the computational and storage efficiencies of generative retrieval with the robust embedding quality and ranking capabilities of dense retrieval. Our SID-based hybrid model applies dense retrieval to a limited set of candidates generated by a generative retrieval module, retaining the minimal storage requirements of generative retrieval while significantly improving performance, particularly for cold-start items. Specifically, our key contributions are as follows:

- We identify and analyze two primary observed limitations of the generative retrieval method on the small-scale academic benchmark: (1) Generative retrieval exhibits a performance difference compared to dense retrieval, given the same item information input, and (2) it tends to overfit to items encountered during training, resulting in a lower probability of generating cold-start items.

- We propose LIGER (LeveragIng dense retrieval for GEnerative Retrieval), a novel method that synergistically combines the strengths of dense and generative retrieval to significantly enhance the performance of generative retrieval. By integrating these methodolo-

gies, LIGER reduces the observed performance differences between dense and generative retrieval while improving the generation of cold-start items on the dataset we explored.

## 2 ANALYSIS OF GENERATIVE AND DENSE RETRIEVAL METHODS

In this section, we first introduce the generative retrieval (Rajput et al., 2024) and dense retrieval (Hou et al., 2022b) formula (see Section 2.1 and 2.2). Then in Section 2.2, we examine the performance difference between generative retrieval and sequential dense retrieval methods, and then discuss the challenges generative retrieval faces in handling item cold-start scenarios in Section 2.4.

### 2.1 GENERATIVE RETRIEVAL REVIEW

The generative retrieval approach such as TIGER (Rajput et al., 2024) typically follows a two-stage training process. The first stage involves collecting textual descriptions for each item based on their attributes. These descriptions serve as inputs to a content model (e.g., a language encoder) that produces item's text embeddings, subsequently quantized by an RQ-VAE (Lee et al., 2022a) to attribute a semantic ID for each item. Formally, for each item $i \in \mathcal{I}$, we collect its $p$ key-value attribute pairs $\{(k_1, v_1), (k_2, v_2), \ldots, (k_p, v_p)\}$ and format them into a textual description: $T_i = \text{prompt}(k_1, v_1, \cdots, k_p, v_p)$. The textual description $T_i$ is then passed to the content model, yielding the text representation $\mathbf{e}_i^{\text{text}}$ for each item. We refer readers to Rajput et al. (2024); Lee et al. (2022a) for the training details of the RQ-VAE module. After the RQ-VAE is trained, we obtain the $m$-tuple semantic ID $(s_i^1, \cdots, s_i^m)$ for each item $i$. Notably, an $m$-tuple semantic ID, with a codebook size $t$, can theoretically represent $t^m$ unique items.

In the second stage of training, the item text representation $\{\mathbf{e}_i^{\text{text}}\}$ and the trained RQ-VAE model are discarded, retaining only the semantic IDs. For each interaction history indexed by item IDs $\{i_1, i_2, \cdots, i_n\}$, the item IDs are replaced with their corresponding semantic IDs: $\{(s_1^1, s_1^2, \cdots, s_1^m), (s_2^1, s_2^2, \cdots, s_2^m), \cdots, (s_n^1, s_n^2, \cdots, s_n^m)\}$. Given the semantic IDs of the last $n$ items a user interacted with, the Transformer model is then optimized to predict the next semantic ID sequence $(s_{n+1}^1, s_{n+1}^2, \cdots, s_{n+1}^m)$.

During inference, a set of candidate items is retrieved using beam search over the trained Transformer, selecting items based on their semantic IDs. A visual representation of the generative retrieval method is provided in Figure 2 *(Lower Left)*.

It is worth noting that although the item text representations are excluded from the second stage of training, they still contribute to the information utilized in developing this approach. To ensure a fair comparison, we incorporate this information into the development of a comparable dense retrieval method in the following section, thereby accounting for the impact of these embeddings on the overall performance.

### 2.2 SEQUENTIAL DENSE RETRIEVAL IN TRANSDUCTIVE SETTING

Sequential dense retrieval methods typically consist of two main components: (1) learning item representations through sequence modeling, and (2) performing retrieval using dot-product search. To enhance the learning of item representations, several dense retrieval methods such as Hou et al. (2022b) employ an transductive setting, where item content information is integrated through text representation to enable transferable representation learning.

Building on these insights, we implement the dense retrieval method as follows: For each item $i$, we first obtain its text representation $\mathbf{e}_i^{\text{text}}$ using the procedure described in the first stage of Section 2.1. Additionally, we retrieve the learnable item embedding $\mathbf{e}_i = \text{Embd}(i)$ from the embedding table $\text{Embd}(\cdot)$ and compute the item's positional embedding $\mathbf{e}_i^{\text{pos}}$. The input embedding for each item is then computed as:

$$\mathbf{E}_i = \mathbf{e}_i + \mathbf{e}_i^{\text{text}} + \mathbf{e}_i^{\text{pos}},$$

and the sequence of input embeddings $\{\mathbf{E}_1, \mathbf{E}_2, \cdots, \mathbf{E}_n\}$ is provided to the Transformer.

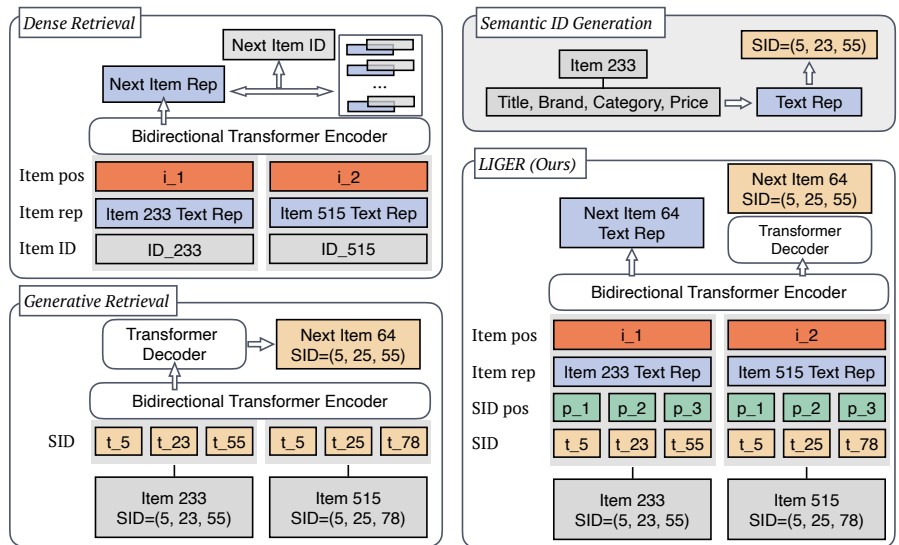

Figure 2: *Overview of Sequential Dense Retrieval, Generative Retrieval, and Our Hybrid Retrieval Method, LIGER. Dense Retrieval (upper left) uses an encoder model to map item IDs and text representations into dense embeddings, which are used to predict the next item in the sequence based on similarity. Generative Retrieval (lower left) employs an encoder-decoder Transformer to generate the next item's semantic ID from the given semantic ID trajectory. These semantic IDs are derived from item features such as title, brand, price, and category (upper right). Our proposed Hybrid Retrieval, LIGER (lower right) combines both semantic ID input and item text representations, integrating dense and generative retrieval techniques. By taking item positions, text representations, and semantic IDs as input, and outputs both the predicted item embedding and the next item's representation.*

Given these $n$ embeddings, the Transformer is trained to optimize the following objective:

$$\mathcal{L}_{\text{dense}}(\Theta, \{\mathbf{e}_i\}, \{\mathbf{e}_i^{\text{pos}}\}) = -\log \frac{\exp\left(\text{sim}(\hat{\mathbf{E}}(\Theta), \mathbf{e}_{n+1} + \mathbf{e}_{n+1}^{\text{text}})/\tau\right)}{\sum_{i\in\mathcal{I}} \exp\left(\text{sim}(\hat{\mathbf{E}}(\Theta), \mathbf{e}_i + \mathbf{e}_i^{\text{text}})/\tau\right)},$$

where $\hat{\mathbf{E}}(\Theta)$ represents the output embedding of the Transformer with parameter $\Theta$, $\text{sim}(\cdot, \cdot)$ denotes the cosine similarity metric, $\tau$ is a temperature scaling factor, and $\mathcal{I}$ is the set of all items. Figure 2 *(Upper Left)* provides a detailed illustration of the dense retrieval model implementation.

## 2.3 THE OBSERVED PERFORMANCE DIFFERENCE

As detailed in Section 2.1, the generative retrieval model leverages both item text representations and sequential interaction information. In Section 2.2, we introduce sequential dense retrieval methods inspired by Hou et al. (2022c), designed to fully incorporate these sources of information throughout the training process. To ensure a fair comparison between the generative retrieval and dense retrieval methods, we maintain consistency in model architecture, data pre-processing, and information utilization. The specific details of our experiment setup are described in Section 4.1 and in Appendix A.2.1. Due to the unavailability of TIGER's codebase, we conducted an extensive hyperparameter search, and configured the dense model's hyperparameters to align with the tuned TIGER settings. However these results, shown in Figure 1 *(Right)*, indicate a performance difference between the generative and dense retrieval methods in the datasets we evaluated.

There are notable differences between the two implemented methods: (1) **Number of Embeddings**: As discussed in Section 2.1, representing $N$ item requires the dense retrieval method to learn and store $\mathcal{O}(N)$ embeddings. In contrast, the semantic-ID-based generative retrieval method only requires $\mathcal{O}(t)$ tokens, where $t^m \approx N$, and $m$ is the length of the semantic ID tuple; (2) **Text Representation Input**: Dense retrieval utilize the item's text representation as additional input; and (3) **Prediction Mechanism**: Dense retrieval relies on maximum inner product search in the embedding space, whereas generative retrieval predicts the next item through next-token prediction via beam search.

To analyze the effect of (1), we modify the dense retrieval approach by replacing item IDs with semantic IDs while keeping all other components unchanged. This modified method is referred to as *Dense (SID)*. Formally, for each item $i$ with semantic ID $(s_i^1, s_i^2, \cdots, s_i^m)$, we construct the input embedding for each semantic ID as:

$$\mathbf{E}_{s_i^j} = \mathbf{e}_{s_i^j} + \mathbf{e}_i^{\texttt{text}} + \mathbf{e}_i^{\texttt{pos}} + \mathbf{e}_j^{\texttt{pos}},$$

where the $\mathbf{e}_{s_i^j}$ is the learnable embedding for each semantic ID: $\mathbf{e}_{s_i^j} = \texttt{Embd}(s_i^j)$, and $\mathbf{e}_j^{\texttt{pos}}$ is the positional embedding for each semantic ID. The final embedding for item i is then represented as: $\mathbf{E}_i = [\mathbf{E}_{s_i^1}, \mathbf{E}_{s_i^2}, \cdots, \mathbf{E}_{s_i^m}]$. During training, the cosine similarity between the predicted embedding and item's text representation is compared and maximized.

To examine the effect of (2), we augment TIGER with the item's text representation, referred to as *TIGER (T)*. Specifically, we use the same input as described earlier and, during training, apply the next-token prediction loss on the semantic ID tuple of the next item.

The results in Figure 3 demonstrate that incorporating semantic IDs as input in dense retrieval (*Dense (SID)*) significantly reduces the performance gap with standard dense retrieval. Moreover, supplementing TIGER with text representation as input (*TIGER (T)*) yields marginal improvements over TIGER alone; however, it still falls short of matching the performance of dense retrieval. This suggests that the primary contributor to the performance gap lies in the inefficiency of the next-token prediction loss in generating retrieved items, rather than the semantic ID representation.

Additionally, when evaluating performance on cold-start item generation (discussed in detail in the next section), both TIGER and TIGER (T) fail, despite including text representation (Hou et al., 2022c; Li et al., 2023b) being known to generalize to cold-start items. Notably, both dense retrieval methods—using either item IDs or semantic IDs—exhibit non-zero performance in cold-start item generation. This highlights a fundamental limitation of generative retrieval methods, where the decoding process itself may impede effective cold-start generation.

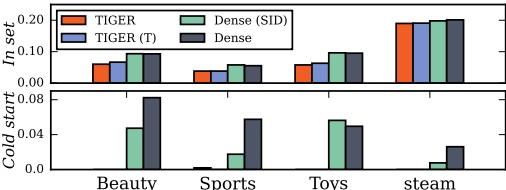

Figure 3: *Comparison of Recall@10 on in-set and cold-start dataset across various datasets.* Dense retrieval methods (both ID- and SID-based) exhibit robust performance in both in-set and cold-start settings. While *TIGER (T)* shows marginal improvements over TIGER, it still lags behind the performance of dense retrieval methods.

### 2.4 CHALLENGES IN COLD-START ITEM PREDICTION WITH GENERATIVE RETRIEVAL MODELS

As partially discussed in the previous section, our investigation extends to the cold-start item generation problem, a critical issue in the dynamic environment of real-world recommendation systems. As new items are continuously introduced, they often lack sufficient user interactions, which impedes their predictability until a significant amount of interaction data is gathered. For dense retrieval in the transductive setting (Hou et al., 2022c), the inclusion of item's text representations provide some prior information, thus partially retaining the ability to retrieve cold-start items, resulting in non-zero performance of cold-start item prediction shown in Figure 1 and Figure 3.

Hence, a natural research question arises: *Can generative retrieval models, which also leverage item's text representations in their process to generate semantic indices, predict cold-start items?* To address this, we analyzed the generation probabilities of cold-start items using a trained generative retrieval model and summarized the results in Figure 4 (b and c). Our findings indicate that the model's learned conditional probabilities tends to overfit to items seen during training, leading to a significantly reduced capability to generate cold-start items. Specifically, we ask the model to generate $K$ candidates using beam search, which ranks items by their generation probabilities. Among these candidates, we define the minimum generation probability as $p_K$. Separately, we calculate the generation probability of the ground-truth cold-start item, denoted as $p^\star$. As shown in Figure 4, the generation probability of cold-start items always falls below the threshold required for inclusion in the beam search process ($p^\star < p_K$), effectively preventing these items from being retrieved. This

limitation highlights the challenges generative retrieval models face in generalizing to unseen items, emphasizing the need for further research and improvements in this area.

It is worth noting that Rajput et al. (2024), propose an alternative solution to mitigate the issue of cold-start item generation. Their approach involves setting a predefined threshold $\varepsilon$ for cold-start item within the retrieved candidate set of $K$ items. This ensures that $K \cdot \varepsilon$ of the retrieved candidates are cold-start items by excluding other candidates that have higher generation probabilities. However, this method relies on prior knowledge of the ratio between recommended cold-start and non-cold-start items, which may not always be available. Therefore, we argue that the challenges in cold-start item generation persist for generative retrieval models, indicating a need for more robust solutions that do not depend heavily on predefined parameters or assumptions.

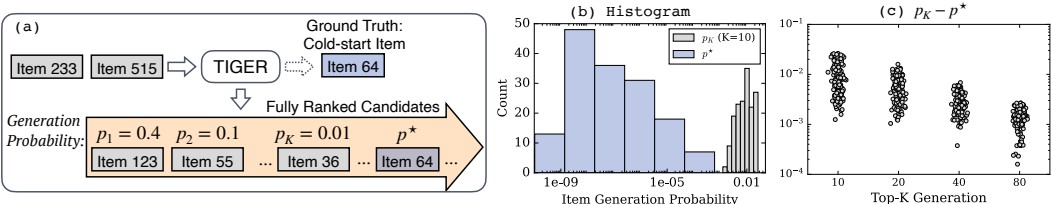

Figure 4: *TIGER Fails to Generate Cold-Start Items.* (a) The TIGER model generates a ranked list of candidates, with $p_K$ denoting the generation probability of generating the $K$-th ranked item over all items. The ground-truth cold-start item has a generation probability of $p^\star$. (b) A histogram compares $p_K$ (for $K = 10$) with $p^\star$ when the ground-truth item is cold-start, highlighting the disparity between them. (c) The difference $p_{\text{diff}} = p_K - p^\star$ is plotted for $K = 10, 20, 40, 80$. A successful generation of cold-start item occurs only when $p_{\text{diff}} \leq 0$, illustrating the model's limitations in handling cold-start items.

## 3 METHODOLOGY

The notion that "there is no free lunch" holds true in the context of retrieval methods. As discussed in the previous section, a performance difference is observed between the generative retrieval and dense retrieval methods we implemented, with generative retrieval facing challenges in generating cold-start items. The improved performance of dense retrieval, however, comes with higher storage, learning, and inference costs. On the other hand, generative retrieval offers efficiency and the ability to generate diverse recommendations (Rajput et al., 2024), but its performance lags behind in the datasets we evaluated.

The trade-offs between approaches are summarized in Table 1, where $N$ represents the total number of items, $t$ denotes the total number of semantic IDs, and $K$ is the number of candidates to be retrieved during inference. Here, we denote the inference cost as the number of comparisons required for each method. It is worth noting that the actual inference time depends on implementation details and infrastructure optimization, which are beyond the scope of this work.

Table 1: *Comparison of Dense Retrieval, Generative Retrieval, and Our Hybrid Retrieval Methods Across Different Costs.* Here $N$ represents the total number of items, $t$ denotes the total number of semantic IDs used by generative retrieval method, and $K$ indicates the number of candidates retrieved during inference.

|  | Dense Retrieval | Generative Retrieval | LIGER (Ours) |
| --- | --- | --- | --- |
| *Learnable Embedding* | $\mathcal{O}(N)$ | $\mathcal{O}(t)$ | $\mathcal{O}(t)$ |
| *(Fixed) Item Text Representation* | $\mathcal{O}(N)$ | - | $\mathcal{O}(N)$ |
| *Inference Cost* | $\mathcal{O}(N)$ | $\mathcal{O}(tK)$ | $\mathcal{O}(tK)$ |
| *Cold-Start Item Generation* | Yes | No | Yes |

In this section, we propose a hybrid method, called LIGER, that combines the strengths of both approaches. Our goal is to improve upon the existing generative retrieval method: enabling it to generate cold-start items and bridging the gap with dense retrieval. To achieve this, we integrate text representations into the sequential model training phase of the generative retrieval method. The associated costs of LIGER are detailed in the last column of Table 1.

Formally, for each item $i$ with semantic ID $(s_i^1, s_i^2, \cdots, s_i^m)$, we construct the input embedding for each semantic ID using the same approach as described in the ablation study in Section 2.2:

$$\mathbf{E}_{s_i^j} = \mathbf{e}_{s_i^j} + \mathbf{e}_i^{\text{text}} + \mathbf{e}_i^{\text{pos}} + \mathbf{e}_j^{\text{pos}},$$

where $\mathbf{e}_{s_i^j}$ is the learnable embedding for the $s_i^j$ semantic ID, $\mathbf{e}_i^{\text{text}}$ is the item's text representation, $\mathbf{e}_i^{\text{pos}}$ is the positional embedding for the item, and $\mathbf{e}_j^{\text{pos}}$ is the positional embedding for the semantic ID. The final embedding for item $i$ is then represented as: $\mathbf{E}_i = [\mathbf{E}_{s_i^1}, \mathbf{E}_{s_i^2}, \cdots, \mathbf{E}_{s_i^m}]$.

During training, the model is trained on two objective: the cosine similarity loss and the next-token prediction loss on the semantic IDs of the next item. The combined loss is formulated as:

$$\mathcal{L}(\Theta, \{\mathbf{e}_i\}, \{\mathbf{e}_i^{\text{pos}}\}, \{\mathbf{e}_j^{\text{pos}}\}) = -\log \frac{\exp\left(\text{sim}(\hat{\mathbf{E}}(\Theta), \mathbf{e}_{n+1}^{\text{text}})/\tau\right)}{\sum_{i \in \mathcal{I}} \exp\left(\text{sim}(\hat{\mathbf{E}}(\Theta), \mathbf{e}_i^{\text{text}})/\tau\right)} - \sum_{j=1}^{m} \log P(s_{n+1}^j \mid [\mathbf{E}_1, \cdots, \mathbf{E}_n]; \Theta).$$

The first term in the loss function ensures that the model learns to align the *encoder*'s output embedding with the text representation of the next item using a softmax over cosine similarity. The second term corresponds to the next-token prediction loss, where each token of the next item's semantic ID tuple is predicted sequentially in the *decoder*, conditioned on the historical input embeddings $[\mathbf{E}_1, \cdots, \mathbf{E}_n]$. Figure 2 (*Lower Right*) provides a detailed illustration of LIGER.

During inference, the decoder retrieves $K$ items by beam search, then supplemented with cold-start items and ranked with the encoder's output embeddings. This design choice is based on the hypothesis that cold-start items are relatively sparse compared to in-set items, as recommendation systems periodically update themselves to incorporate newly introduced items. Therefore, we supplement the candidates with cold-start items to ensure their inclusion, as their number is relatively small. Details are shown in Figure 5 (*left*).

---

**Algorithm 1:** Inference Process

**Input** : Interaction sequence $\{\mathbf{E}_1, \mathbf{E}_2, \ldots, \mathbf{E}_n\}$, Cold-start items $\mathcal{C}$, Beam size $K$

**Output:** Ranked list of items $\hat{\mathcal{I}}$

1. Beam search to retrieve top-$K$ candidates:
   $\mathcal{I}_{\text{beam}} = \text{TF}([\mathbf{E}_1, \mathbf{E}_2, \ldots, \mathbf{E}_n]; K)$;
2. Combine with cold-start items:
   $\mathcal{I}_{\text{comb}} = \mathcal{I}_{\text{beam}} \cup \mathcal{C}$;
3. Rank Candidates with encoder's output $\hat{\mathbf{E}}$:
   $\hat{\mathcal{I}} = \text{topk}(\text{sim}(\hat{\mathbf{E}}, \mathbf{e}_i^{\text{text}}), \quad \forall i \in \mathcal{I}_{\text{comb}})$;

---

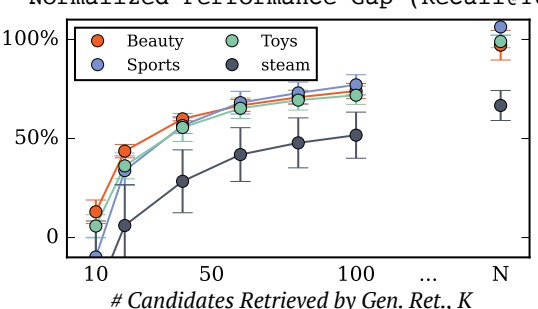

Figure 5: *Inference Process and* LIGER*'s Performance in Bridging the Gap as the Number of Retrieved Candidates Increases.* The left panel illustrates the inference process of LIGER, detailing how candidate items are retrieved and ranked. In the algorithm, $\text{TF}(\cdot, K)$ denotes the Transformer generating $K$ candidates using beam search based on the input sequence. The right panel shows the normalized performance gap between generative and dense retrieval models across several datasets (Beauty, Sports, Toys, Steam) for the in-set Recall@10 metric. In this normalization, 0% represents the performance of the generative retrieval model, while 100% corresponds to the performance of the dense retrieval model. The figure highlights how LIGER progressively bridges the performance gap as the number of candidates retrieved by the generative model increases.

To demonstrate the efficacy of LIGER, we evaluate how its performance varies with the number of candidates K retrieved by the generative retrieval. For clearer insights, we present the normalized performance gap (NPG) in Recall@10 across various datasets. Specifically, let the performance of LIGER with $K$ candidates retrieved by generative retrieval be $r(K)$, the performance of TIGER be $r_{\text{TIGER}}$, and the performance of dense retrieval be $r_{\text{dense}}$. The NPG is then defined as:

$$\text{NPG}(K) = {(r(K) - r_{\text{TIGER}})}/{(r_{\text{dense}} - r_{\text{TIGER}})}.$$

We plot the NPG values for each dataset in Figure 5 (*right*) with varying $K$. The results show a consistent interpolation between TIGER and the dense retrieval approach: as the number of candidates $K$ retrieved by the generative model increases, the likelihood of including the correct items, along with the cold-start candidates, in the candidate set grows. Consequently, LIGER progressively improves its performance on the Recall@10 metric, narrowing the gap with the dense retrieval method. In the next section, we will demonstrate the effectiveness of our method across various datasets and baseline methods.

# 4 EXPERIMENTAL SETUP AND RESULTS

In this section, we present the experimental results across various datasets and baseline methods, showcasing the performance on both in-set and cold-start items. Specifically, we assess the cold-start performance by testing on items that are unseen during training, which is determined by the dataset statistics.

## 4.1 EXPERIMENTAL SETUP

**Datasets**. We evaluate LIGER on four datasets, preprocessing them using the standard 5-core filtering method (Zhang et al., 2019; Zhou et al., 2020). This process removes items with fewer than 5 users and users with fewer than 5 interactions. Additionally, we truncate sequences to a maximum length of 20, retaining the most recent items. Detailed statistics of the resulting datasets are provided in Appendix A.2.3.

• *Amazon Beauty, Sports, and Toys* (He & McAuley, 2016): We use the Amazon Review dataset (2014), focusing on three categories: `Beauty`, `Sports and Outdoors`, and `Toys and Games`. For each item, we construct embeddings by incorporating four key attributes: title, price, category, and description.

• *Steam* (Kang & McAuley, 2018b): The dataset comprises online reviews of video games, from which we extract relevant attributes to construct item embeddings. Specifically, we utilize the following attributes: title, genre, specs, tags, price, and publisher. To reduce the dataset size and make it more manageable, we apply subsampling by selecting every 7th sequence, thereby retaining a representative subset of the data.

When generating the item text representations, the item attributes are processed using the sentence-T5 model Ni et al. (2021) (XXL).

**Semantic ID Generation**. Utilizing the text representations generated from the sentence-T5 model, we employ a 3-layer MLP for both the encoder and decoder in the RQ-VAE Lee et al. (2022a). The RQ-VAE features three levels of learnable codebooks, each with a dimension of 128 and a cardinality of 256. We use the AdamW optimizer to train the RQ-VAE, setting the learning rate at 0.001 and the weight decay at 0.1. To prevent collisions (i.e., the same semantic ID representing different items), following Rajput et al. (2024) we append an extra token at the end of the ordered semantic codes to ensure uniqueness.

**Sequential Modeling Architecture and Training Algorithm**. For the generative model, we utilize the T5 (Raffel et al., 2020) encoder-decoder model, configuring both the encoder and decoder with 6 layers, an embedding dimension of 128, 6 heads, and a feed-forward network hidden dimension of 1024. The dropout rate is 0.2. The dense retrieval model designed in Section 2.2 employs only the T5-encoder with 6 layers, while maintaining the same hyper-parameters. We use the AdamW optimizer with a learning rate of 0.0003, a weight decay parameter of 0.035, and a cosine learning rate scheduler. Additional details are presented in Appendix A.2.1.

**Evaluation Metrics**. We assess the model's performance using Normalized Discounted Cumulative Gain (NDCG)@10 and Recall@10. For dataset splitting, we adopt the leave-one-out strategy following (Kang & McAuley, 2018b; Zhou et al., 2020; Rajput et al., 2024), designating the last item as the test label, the preceding item for validation, and the remainder for training. During training, early stopping is applied based on the in-set NDCG@10 validation metric. For LIGER, which comprises two components: (A) the semantic ID prediction head and (B) the output embedding head, we implement early stopping based on the performance of component (B). To ensure fair evaluation of cold-start items, we exclude them from the RQ-VAE training to avoid data contamination.

**Baselines**. We compare our method against five state-of-the-art Item-ID-based dense retrieval methods: (a) SASRec (Kang & McAuley, 2018b); feature-informed methods: (b) FDSA (Zhang et al., 2019), (c) $S^3$-Rec (Zhou et al., 2020); and modality-based methods: (d) UnisRec (Hou et al., 2022b), (e) Recformer (Li et al., 2023a). Descriptions and implementation details for these baselines are provided in Appendix A.2.4. We also compare LIGER against TIGER (Rajput et al., 2024), a semantic ID-based generative retrieval method. Although subsequent works have built upon this paradigm using large language models (LLMs) (Zheng et al., 2023; Cao et al., 2024b), they rely on pre-trained LLMs, which are outside the scope of our comparisons.

Table 2: *Performance Comparison Across Baseline Methods on Amazon Beauty, Sports, Toys, and Steam Datasets.* The best performance is highlighted in bold, and the second-best performance is underlined. Our method consistently achieves either the best or second-best performance across all datasets, closely followed by modality-based baselines (UniSRec or RecFormer). We report LIGER's results where generative retrieval is used to retrieve 20 items, followed by the sequential dense retrieval.

| | Methods | Inference Cost | NDCG@10↑ | | Recall@10↑ | |
| | | | In-set | Cold | In-set | Cold |
|---|---|---|---|---|---|---|
| Beauty | SASRec | $\mathcal{O}(N)$ | $0.02179 \pm 0.00023$ | $0.0 \pm 0.0$ | $0.05109 \pm 0.00042$ | $0.0 \pm 0.0$ |
| | FDSA | $\mathcal{O}(N)$ | $0.02244 \pm 0.00135$ | $0.0 \pm 0.0$ | $0.04530 \pm 0.00357$ | $0.0 \pm 0.0$ |
| | S³-Rec | $\mathcal{O}(N)$ | $0.02279 \pm 0.00058$ | $0.0 \pm 0.0$ | $0.05226 \pm 0.00229$ | $0.0 \pm 0.0$ |
| | UniSRec | $\mathcal{O}(N)$ | $\underline{0.03346 \pm 0.00057}$ | $0.01422 \pm 0.00128$ | $\underline{0.06937 \pm 0.00110}$ | $0.03704 \pm 0.00000$ |
| | RecFormer | $\mathcal{O}(N)$ | $0.02880 \pm 0.00085$ | $\underline{0.01955 \pm 0.00433}$ | $0.06265 \pm 0.00196$ | $\underline{0.04733 \pm 0.00943}$ |
| | TIGER | $\mathcal{O}(tK)$ | $0.03216 \pm 0.00084$ | $0.0 \pm 0.0$ | $0.06009 \pm 0.00204$ | $0.0 \pm 0.0$ |
| | LIGER (Ours) | $\mathcal{O}(tK)$ | $\mathbf{0.04020 \pm 0.00044}$ | $\mathbf{0.03800 \pm 0.00523}$ | $\mathbf{0.07447 \pm 0.00111}$ | $\mathbf{0.10082 \pm 0.01285}$ |
| Sports | SASRec | $\mathcal{O}(N)$ | $0.01160 \pm 0.00038$ | $0.0 \pm 0.0$ | $0.02696 \pm 0.00102$ | $0.0 \pm 0.0$ |
| | FDSA | $\mathcal{O}(N)$ | $0.01391 \pm 0.00162$ | $0.0 \pm 0.0$ | $0.02699 \pm 0.00312$ | $0.0 \pm 0.0$ |
| | S³-Rec | $\mathcal{O}(N)$ | $0.01097 \pm 0.00033$ | $0.0 \pm 0.0$ | $0.02557 \pm 0.00034$ | $0.0 \pm 0.0$ |
| | UniSRec | $\mathcal{O}(N)$ | $0.01814 \pm 0.00041$ | $0.00676 \pm 0.00244$ | $0.03753 \pm 0.00106$ | $0.01559 \pm 0.00447$ |
| | RecFormer | $\mathcal{O}(N)$ | $0.01318 \pm 0.00053$ | $\underline{0.01797 \pm 0.00000}$ | $0.02921 \pm 0.00167$ | $\underline{0.03801 \pm 0.00000}$ |
| | TIGER | $\mathcal{O}(tK)$ | $\underline{0.01989 \pm 0.00085}$ | $0.00064 \pm 0.00056$ | $\underline{0.03822 \pm 0.00109}$ | $0.00195 \pm 0.00169$ |
| | LIGER (Ours) | $\mathcal{O}(tK)$ | $\mathbf{0.02430 \pm 0.00075}$ | $\mathbf{0.02731 \pm 0.00229}$ | $\mathbf{0.04400 \pm 0.00127}$ | $\mathbf{0.05848 \pm 0.00292}$ |
| Toys | SASRec | $\mathcal{O}(N)$ | $0.02756 \pm 0.00079$ | $0.0 \pm 0.0$ | $0.06314 \pm 0.00178$ | $0.0 \pm 0.0$ |
| | FDSA | $\mathcal{O}(N)$ | $0.02375 \pm 0.00277$ | $0.0 \pm 0.0$ | $0.04684 \pm 0.00483$ | $0.0 \pm 0.0$ |
| | S³-Rec | $\mathcal{O}(N)$ | $0.02942 \pm 0.00071$ | $0.0 \pm 0.0$ | $0.06659 \pm 0.00135$ | $0.0 \pm 0.0$ |
| | UniSRec | $\mathcal{O}(N)$ | $0.03622 \pm 0.00056$ | $0.01090 \pm 0.00084$ | $0.07472 \pm 0.00058$ | $0.02477 \pm 0.00195$ |
| | RecFormer | $\mathcal{O}(N)$ | $\underline{0.03697 \pm 0.00052}$ | $\underline{0.04432 \pm 0.00094}$ | $\mathbf{0.07971 \pm 0.00170}$ | $\underline{0.10023 \pm 0.00516}$ |
| | TIGER | $\mathcal{O}(tK)$ | $0.02949 \pm 0.00049$ | $0.0 \pm 0.0$ | $0.05782 \pm 0.00163$ | $0.0 \pm 0.0$ |
| | LIGER (Ours) | $\mathcal{O}(tK)$ | $\mathbf{0.03756 \pm 0.00151}$ | $\mathbf{0.05231 \pm 0.00531}$ | $\underline{0.07135 \pm 0.00244}$ | $\mathbf{0.13063 \pm 0.00516}$ |
| Steam | SASRec | $\mathcal{O}(N)$ | $0.14763 \pm 0.00051$ | $0.0 \pm 0.0$ | $0.18259 \pm 0.00055$ | $0.0 \pm 0.0$ |
| | FDSA | $\mathcal{O}(N)$ | $0.08236 \pm 0.00152$ | $0.0 \pm 0.0$ | $0.14773 \pm 0.00234$ | $0.0 \pm 0.0$ |
| | S³-Rec | $\mathcal{O}(N)$ | $0.14437 \pm 0.00127$ | $0.0 \pm 0.0$ | $0.18025 \pm 0.00222$ | $0.0 \pm 0.0$ |
| | UniSRec | $\mathcal{O}(N)$ | - | - | - | - |
| | RecFormer | $\mathcal{O}(N)$ | $0.14034 \pm 0.00123$ | $\underline{0.00120 \pm 0.00037}$ | $0.17042 \pm 0.00275$ | $\underline{0.00319 \pm 0.0011}$ |
| | TIGER | $\mathcal{O}(tK)$ | $\mathbf{0.15034 \pm 0.00064}$ | $0.0 \pm 0.0$ | $\underline{0.18980 \pm 0.00135}$ | $0.0 \pm 0.0$ |
| | LIGER (Ours) | $\mathcal{O}(tK)$ | $\underline{0.14951 \pm 0.00158}$ | $\mathbf{0.00512 \pm 0.00047}$ | $\mathbf{0.19049 \pm 0.00234}$ | $\mathbf{0.01466 \pm 0.0011}$ |

**Experimental Results**. The results from the benchmark datasets are presented in Table 2, where the mean and standard deviation are calculated across three random seed runs. Traditional item-ID-based methods, such as SASRec exhibit poor in-set performance compared to semantic-ID-based models. However, when attribute information is included, models like FDSA and S³-Rec show improved in-set performance. Nevertheless, their performance on cold-start items remains subpar due to the static nature of item embeddings. In contrast, models that utilize text representations and pre-training, such as UniSRec and RecFormer, demonstrate enhanced capabilities in handling cold-start item scenarios. The inclusion of text embeddings during pre-training enables these models to better handle unseen items. TIGER, which is a semantic-ID-based generative retrieval model, outperforms item-ID-based methods in terms of in-set performance but still struggles with cold-start item generation.

Our model, LIGER, builds upon TIGER by using semantic-ID-based inputs and combining dense retrieval with semantic ID generation as outputs. This approach significantly improves upon the TIGER method and enables effective generation of cold-start items. Across all datasets, our method consistently achieves either the best or second-best performance, closely followed by modality-based baselines such as UniSRec and RecFormer. We adopt a hybrid approach for our reporting, where we use generative retrieval to retrieve 20 items and then rank them with cold-start items using dense retrieval. Comprehensive result including performance of LIGER with different number of retrieved items from generative retrieval is presented in Table 4. Additional ablation study on each component of LIGER is present in Appendix A.4.

## 5   DISCUSSION

**Addressing Cold-Start Items with Hybrid Retrieval Models.** That generative retrieval method's struggle with cold-start items primarily stems from overfitting to familiar semantic IDs during training, as discussed in Section 2.4. To mitigate this issue, LIGER efficiently combines dense retrieval

with generative retrieval. Specifically, LIGER first generates a small set of $K$ candidates (where $K \ll N$) using generative retrieval, which is then augmented with the cold-start item set. This approach leverages the efficiency of generative retrieval to reduce the candidate pool while ensuring cold-start items are represented. The dense retrieval component further enhances cold-start performance by leveraging item text embeddings as prior information. As shown in Table 4, this integration ensures that when generative retrieval retrieves fewer than N items, the model maintains robust performance in cold-start scenarios, comparable to dense retrieval only.

**Comparative Performance with Current Dense Retrieval Methods.** While LIGER demonstrates competitive performance against existing baselines, its primary objective is to strike a balance between the generative and dense retrieval frameworks. As discussed in previous sections, there are observed performance differences between these two methods on the dataset we tested, even when using the same input information and model architecture. The results presented in this work aim to shed light on potential future directions for integrating these approaches, paving the way for the development of more robust and efficient recommendation systems.

**Observed Performance Differences are Contextual to Small-Scale Datasets.** We want to note that the performance differences observed in this study are influenced by various factors, including dataset size, implementation details, and the distribution of data collected under specific paradigms. Additionally, we are aware of industry-scale implementations of generative retrieval paradigms (Singh et al., 2023) that outperform dense retrieval approaches in real-world settings. In this work, our goal is not to assert a definitive performance gap between the two paradigms. Instead, we focused on aligning the two methods as closely as possible within the scope of academic benchmarks, which typically use small-scale datasets. Our findings are meant to provide insights specific to this academic setting and should not be extrapolated to large-scale, real-world applications without further investigation. We hope that these observations encourage a critical evaluation of academic benchmark design and inspire future research to explore performance under more diverse and realistic conditions, while addressing the limitations of small-scale datasets commonly used in academic studies.

## 6 CONCLUSION

In this work, we conducted a comprehensive comparison between dense retrieval methods and the emerging generative retrieval approach. Our analysis revealed the limitations of dense retrieval, including high computational and storage requirements, while highlighting the advantages of generative retrieval, which uses semantic IDs and generative models to enhance efficiency and semantic understanding. Furthermore, we have identified the challenges faced by generative retrieval, particularly in handling cold-start items and matching the performance of dense retrieval. To address these challenges, we introduced a novel hybrid model, LIGER, that combines the strengths of both approaches. Our findings demonstrate that our hybrid model surpasses existing models in handling cold-start scenarios and achieves advanced overall performance on benchmark datasets.

Looking ahead, the fusion of dense and generative retrieval methods holds tremendous potential for advancing recommendation systems. Our research provides a foundation for further exploration into hybrid models that capitalize on the strengths of both retrieval types. As these models continue to evolve, they will become increasingly practical for real-world applications, enabling more personalized and responsive user experiences.

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

# A APPENDIX

## A.1 RELATED WORK

Due to the page limit, we present the related work here in the Appendix.

**Generative Retrieval.** The concept of generative retrieval was first proposed by Tay et al. (2022) within the domain of document retrieval. This paradigm shifts from traditional search and retrieval methods by encoding document information directly into the weights of a Transformer model. Subsequent studies (De Cao et al., 2020; Bevilacqua et al., 2022; Feng et al., 2022) have expanded on this foundation, enhancing document retrieval through improvements in indexing (Lee et al., 2022b;c; Wang et al., 2022), and the efficient continual database updates (Mehta et al., 2022; Kishore et al., 2023; Chen et al., 2023).

In the realm of sequential recommendation systems, Rajput et al. (2024) is the first work to leverage the generative retrieval techniques. The target item is directly generated given a user's interaction history, rather than selecting top items by ranking all relevant user-item pairs. A key challenge in generative retrieval is striking a balance between memorization and generalization when encoding items. To address this, semantic IDs have been proposed by leveraging RQ-VAE models (Lee et al., 2022a; Van Den Oord et al., 2017). These models encode content-based embeddings into a compact, discrete semantic indexer that captures the hierarchical structure of concepts within an item's content, proving to be scalable in industrial applications (Singh et al., 2023). Recent developments (Hou et al., 2022a) have expanded semantic-ID-based generative retrieval to include contrastive learning (Jin et al., 2024), multimodal integration (Liu et al., 2024a), tokenization techniques (Sun et al., 2024), and learning-to-rank methods (Li et al., 2024).

**Sequential Dense Recommendation.** Traditional sequential dense recommender models follow the paradigm of learning representations of users, items, and their interactions with multimodal data. Early work (Hidasi et al., 2015) proposed architectures based on traditional Recurrent Neural Networks (RNNs), while later studies (Kang & McAuley, 2018a; Sun et al., 2019; de Souza Pereira Moreira et al., 2021) have shifted towards the Transformer architecture to enhance performance. Besides capturing the user-item interaction history pattern with the sequential modeling, extra features such as item attributes (Zhang et al., 2019; Zhou et al., 2020) has been utilized to further improve the performance. With the recent advancements in Large Language Models (LLMs), several works have explored using these models as the backbone for recommender systems, aligning item representations with LLMs to improve recommendation performance (Li et al., 2023b; Hou et al., 2022c; Cao et al., 2024a; Zheng et al., 2024). In this work, we aim to merge the sequential dense recommendation approach with generative retrieval techniques, assessing performance gaps and computational costs, and proposing a hybrid method that combines the strengths of both paradigms.

**Cold-start Problem.** Traditional challenges such as long-tail and cold-start items continue to hinder recommendation systems. The long-tail items issue arises from skewed distributions where a few popular items dominate user interactions (Zhang et al., 2022; 2020), while the cold-start problem arises when new items are introduced without any historical interaction data. Recent studies (Hou et al., 2022c; Li et al., 2023b) have shown that textual embeddings can provide a robust prior for tackling the cold-start issue, and further improvements have been achieved by integrating pretrained LLMs (Huang et al., 2024; Sanner et al., 2023) and knowledge graphs (Frej et al., 2024). In this work, we explore the cold-start problem within the context of generative retrieval and propose a hybrid method that combines dense retrieval with textual embeddings to effectively mitigate this issue.

## A.2 EXPERIMENTAL DETAILS

Table 3: Dataset statistics after applying 5-core filtering to both users and items. The first three datasets (Beauty, Sports, and Toys) are subsets of the Amazon review dataset.

| Dataset | # users | # items | # actions | # cold-start items |
|---------|---------|---------|-----------|--------------------|
| *Beauty* | 22,363 | 12,101 | 198,502 | 43 |
| *Toys and Games* | 19,412 | 11,924 | 167,597 | 56 |
| *Sports and Outdoors* | 35,598 | 18,357 | 296,337 | 81 |
| *Steam* | 47,761 | 12,012 | 599,620 | 400 |

### A.2.1 IMPLEMENTATION DETAILS OF DENSE RETRIEVAL

In Section 4.1, we provide the TIGER implementation details. Here, we describe the implementation details of the dense retrieval method we developed.

The dense retrieval method uses the same T5-encoder architecture, configured with 6 layers, an embedding dimension of 128, 6 attention heads, and a feed-forward network hidden dimension of 1024. For content embeddings, we use the same model as TIGER, sentence-T5-XXL, which generates text embeddings in $\mathbf{R}^{768}$. To integrate these embeddings into the Transformer, we project them down to a 128-dimensional space using a linear layer.

Following the notation in Section 2.2, once $\mathbf{E}_i$ is collected for each item, it is passed through a `LayerNorm` layer, followed by a `Dropout` layer with a rate of 0.5. For the cosine similarity loss calculation, we set the temperature parameter $\tau = 0.07$. During training, we use the same learning rate (0.0003), cosine learning rate scheduler, optimizer (AdamW), and weight decay (0.035) as employed in TIGER's training.

### A.2.2 IMPLEMENTATION DIFFERENCE BETWEEN UNISREC AND OUR DENSE MODEL

Although the dense retrieval model we implemented is inspired by UniSRec Hou et al. (2022c), we made several modifications to simplify the model and align it with the best-tuned TIGER architecture. Specifically:

1. We use the same content model as TIGER: T5-XXL, whereas UniSRec uses BERT.

2. We adopt the same encoder architecture as TIGER, which consists of 6 T5 encoder layers with 6 attention heads, an embedding dimension of 128, and a feed-forward network hidden dimension of 1024. In contrast, UniSRec uses a custom Transformer block with 2 layers, 2 attention heads, an embedding size of 300, a hidden size of 256, and different implementations for `LayerNorm` and positional embeddings compared to the T5 model [1].

3. We replace UniSRec's mixture-of-expert layer and whitening layer with a simple `Linear` layer.

4. We train the dense retrieval model from scratch, whereas UniSRec relies on pretraining using the Amazon 2018 datasets.

### A.2.3 DATA STATISTICS

In Table 3, we present the statistics of the datasets used in our evaluation.

### A.2.4 BASELINES

We compare our methods with five state-of-the-art Item-ID-based dense retrieval methods, including:

1. SASRec (Kang & McAuley, 2018b). A self-attention based sequential recommendation model that learns to predict the next item ID based on the user's interaction history.

---

[1] https://github.com/RUCAIBox/UniSRec/blob/master/props/UniSRec.yaml

2. FDSA (Zhang et al., 2019) [*feature-informed*]. This method extends SASRec by incorporating item features into the self-attention model, allowing it to leverage prior information about cold-start items through their attributes.

3. S$^3$-Rec (Zhou et al., 2020) [*feature-informed*]. A self-attention based model that utilizes data correlation to create self-supervision signals, improving sequential recommendation through pre-training.

4. UnisRec (Hou et al., 2022b) [*modality-based*]. A model that learns universal item representations by utilizing associated description text and a lightweight encoding architecture that incorporates parametric whitening and a mixture-of-experts adaptor. We fine-tune the released pretrained model in the transductive setting.

5. Recformer (Li et al., 2023a) [*modality-based*]. A bidirectional Transformer-based model that encodes item information using key-value attributes described by text. We fine-tune the pre-trained model on the downstream datasets.

## A.3 FULL EXPERIMENTAL RESULT

In Table 4, we present the full results on the benchmark, where our method with different number of retrieved candidates from generative retrieval are shown.

## A.4 ABLATION STUDY

LIGER combines dense retrieval and generative retrieval paradigms into a unified framework. As described in Section 3 and illustrated in Figure 2, for each item $i$ with semantic ID $(s_i^1, s_i^2, \cdots, s_i^m)$, we construct the input embedding for each semantic ID as:

$$\mathbf{E}_{s_i^j} = \mathbf{e}_{s_i^j} + \mathbf{e}_i^{\text{text}} + \mathbf{e}_i^{\text{pos}} + \mathbf{e}_j^{\text{pos}},$$

where $\mathbf{e}_{s_i^j}$ is the learnable embedding for the $s_i^j$ semantic ID, $\mathbf{e}_i^{\text{text}}$ is the item's text representation, $\mathbf{e}_i^{\text{pos}}$ is the positional embedding for the item, and $\mathbf{e}_j^{\text{pos}}$ is the positional embedding for the semantic ID. The final embedding for item $i$ is then represented as: $\mathbf{E}_i = [\mathbf{E}_{s_i^1}, \mathbf{E}_{s_i^2}, \cdots, \mathbf{E}_{s_i^m}]$.

During training, the model is optimized with two objectives: the cosine similarity loss and the next-token prediction loss on the semantic IDs of the next item: The cosine similarity loss ensures that the model learns to align the *encoder*'s output embedding with the text representation of the next item, and the next-token prediction loss supervise on the next item's semantic ID tuple prediction performance. To summarize, the LIGER framework is structured as follows:

1. *Input:* Semantic ID (SID) and item's text representation;
2. *Output*: Predictions for:
   (a) The next item's semantic ID through the SID head.
   (b) The next item's text representation through the embedding head.

The ablation study investigates the impact of each component, as shown in Figure 6:

- *LIGER (detach)* detaches the gradient updates from the SID head in LIGER to examine the importance of multi-objective optimization through the SID head;
- *TIGER (T)* removes the embedding head from LIGER, focusing solely on the SID head and the text representation as input;
- *TIGER* further simplifies *TIGER (T)* by removing the item text representation input, reducing the model to the generative retrieval method described in Section 2.1;
- *Dense (SID)* removes the SID head from LIGER, retaining only the dense retrieval mechanism with SID as input;
- *Dense* replaces the SID input with ID in *Dense (SID)*, reducing the model to the dense retrieval method in transductive setting, as detailed in Section 2.2.

Table 4: *Performance Table for Amazon Beauty, Sports, Toys, and Steam Datasets Across Various Baseline Methods.* In this table, we present our method with different number of retrieved candidates $K$ from generative retrieval.

| Datasets | Methods | Inference Cost | NDCG@10↑ | | Recall@10↑ | |
|---|---|---|---|---|---|---|
| | | | In-set | Cold | In-set | Cold |
| Beauty | SASRec | $\mathcal{O}(N)$ | $0.02179 \pm 0.00023$ | $0.0 \pm 0.0$ | $0.05109 \pm 0.00042$ | $0.0 \pm 0.0$ |
| | FDSA | $\mathcal{O}(N)$ | $0.02244 \pm 0.00135$ | $0.0 \pm 0.0$ | $0.04530 \pm 0.00357$ | $0.0 \pm 0.0$ |
| | S$^3$-Rec | $\mathcal{O}(N)$ | $0.02279 \pm 0.00058$ | $0.0 \pm 0.0$ | $0.05226 \pm 0.00229$ | $0.0 \pm 0.0$ |
| | UniSRec | $\mathcal{O}(N)$ | $0.03346 \pm 0.00057$ | $0.01422 \pm 0.00128$ | $0.06937 \pm 0.00110$ | $0.03704 \pm 0.00000$ |
| | RecFormer | $\mathcal{O}(N)$ | $0.02880 \pm 0.00085$ | $0.01955 \pm 0.00433$ | $0.06265 \pm 0.00196$ | $0.04733 \pm 0.00943$ |
| | TIGER | $\mathcal{O}(tK)$ | $0.03216 \pm 0.00084$ | $0.0 \pm 0.0$ | $0.06009 \pm 0.00204$ | $0.0 \pm 0.0$ |
| | Ours ($K=20$) | $\mathcal{O}(tK)$ | $0.0402 \pm 0.00044$ | $0.038 \pm 0.00523$ | $0.07447 \pm 0.00111$ | $0.10082 \pm 0.01285$ |
| | Ours ($K=40$) | $\mathcal{O}(tK)$ | $0.0423 \pm 0.00067$ | $0.02815 \pm 0.00525$ | $0.07985 \pm 0.00033$ | $0.07613 \pm 0.01285$ |
| | Ours ($K=60$) | $\mathcal{O}(tK)$ | $0.04328 \pm 0.00113$ | $0.02609 \pm 0.00343$ | $0.08207 \pm 0.00101$ | $0.07202 \pm 0.00943$ |
| | Ours ($K=80$) | $\mathcal{O}(tK)$ | $0.04392 \pm 0.00097$ | $0.0255 \pm 0.00425$ | $0.08343 \pm 0.00115$ | $0.07202 \pm 0.00943$ |
| | Ours ($K=100$) | $\mathcal{O}(tK)$ | $0.0443 \pm 0.00101$ | $0.02469 \pm 0.00381$ | $0.08447 \pm 0.00126$ | $0.07202 \pm 0.00943$ |
| | Ours ($K=N$) | $\mathcal{O}(N)$ | $0.04738 \pm 0.00151$ | $0.01225 \pm 0.00256$ | $0.0921 \pm 0.00247$ | $0.03704 \pm 0.00617$ |
| Sports | SASRec | $\mathcal{O}(N)$ | $0.01160 \pm 0.00038$ | $0.0 \pm 0.0$ | $0.02696 \pm 0.00102$ | $0.0 \pm 0.0$ |
| | FDSA | $\mathcal{O}(N)$ | $0.01391 \pm 0.00162$ | $0.0 \pm 0.0$ | $0.02699 \pm 0.00312$ | $0.0 \pm 0.0$ |
| | S$^3$-Rec | $\mathcal{O}(N)$ | $0.01097 \pm 0.00033$ | $0.0 \pm 0.0$ | $0.02557 \pm 0.00034$ | $0.0 \pm 0.0$ |
| | UniSRec | $\mathcal{O}(N)$ | $0.01814 \pm 0.00041$ | $0.00676 \pm 0.00244$ | $0.03753 \pm 0.00106$ | $0.01559 \pm 0.00447$ |
| | RecFormer | $\mathcal{O}(N)$ | $0.01318 \pm 0.00053$ | $0.01797 \pm 0.00000$ | $0.02921 \pm 0.00167$ | $0.03801 \pm 0.00000$ |
| | TIGER | $\mathcal{O}(tK)$ | $0.01989 \pm 0.00085$ | $0.00064 \pm 0.00056$ | $0.03822 \pm 0.00109$ | $0.00195 \pm 0.00169$ |
| | Ours ($K=20$) | $\mathcal{O}(tK)$ | $0.0243 \pm 0.00075$ | $0.02731 \pm 0.00229$ | $0.044 \pm 0.00127$ | $0.05848 \pm 0.00292$ |
| | Ours ($K=40$) | $\mathcal{O}(tK)$ | $0.02594 \pm 0.00063$ | $0.01814 \pm 0.00052$ | $0.04789 \pm 0.00067$ | $0.04191 \pm 0.00338$ |
| | Ours ($K=60$) | $\mathcal{O}(tK)$ | $0.02672 \pm 0.00067$ | $0.01517 \pm 0.00068$ | $0.04987 \pm 0.00098$ | $0.03411 \pm 0.00338$ |
| | Ours ($K=80$) | $\mathcal{O}(tK)$ | $0.02714 \pm 0.00064$ | $0.0127 \pm 0.00096$ | $0.05071 \pm 0.00094$ | $0.02924 \pm 0.00506$ |
| | Ours ($K=100$) | $\mathcal{O}(tK)$ | $0.02744 \pm 0.00055$ | $0.01175 \pm 0.00059$ | $0.05141 \pm 0.00087$ | $0.02729 \pm 0.00169$ |
| | Ours ($K=N$) | $\mathcal{O}(N)$ | $0.02962 \pm 0.00053$ | $0.00581 \pm 0.00238$ | $0.05641 \pm 0.00071$ | $0.01365 \pm 0.00447$ |
| Toys | SASRec | $\mathcal{O}(N)$ | $0.02756 \pm 0.00079$ | $0.0 \pm 0.0$ | $0.06314 \pm 0.00178$ | $0.0 \pm 0.0$ |
| | FDSA | $\mathcal{O}(N)$ | $0.02375 \pm 0.00277$ | $0.0 \pm 0.0$ | $0.04684 \pm 0.00483$ | $0.0 \pm 0.0$ |
| | S$^3$-Rec | $\mathcal{O}(N)$ | $0.02942 \pm 0.00071$ | $0.0 \pm 0.0$ | $0.06659 \pm 0.00135$ | $0.0 \pm 0.0$ |
| | UniSRec | $\mathcal{O}(N)$ | $0.03622 \pm 0.00056$ | $0.01090 \pm 0.00084$ | $0.07472 \pm 0.00058$ | $0.02477 \pm 0.00195$ |
| | RecFormer | $\mathcal{O}(N)$ | $0.03697 \pm 0.00052$ | $0.04432 \pm 0.00094$ | $0.07971 \pm 0.00170$ | $0.10023 \pm 0.00516$ |
| | TIGER | $\mathcal{O}(tK)$ | $0.02949 \pm 0.00049$ | $0.0 \pm 0.0$ | $0.05782 \pm 0.00163$ | $0.0 \pm 0.0$ |
| | Ours ($K=20$) | $\mathcal{O}(tK)$ | $0.03756 \pm 0.00151$ | $0.05231 \pm 0.00531$ | $0.07135 \pm 0.00244$ | $0.13063 \pm 0.00516$ |
| | Ours ($K=40$) | $\mathcal{O}(tK)$ | $0.04021 \pm 0.00157$ | $0.03574 \pm 0.00262$ | $0.07859 \pm 0.00264$ | $0.09459 \pm 0.00585$ |
| | Ours ($K=60$) | $\mathcal{O}(tK)$ | $0.04163 \pm 0.0014$ | $0.03173 \pm 0.00149$ | $0.08222 \pm 0.00194$ | $0.08559 \pm 0.00516$ |
| | Ours ($K=80$) | $\mathcal{O}(tK)$ | $0.04245 \pm 0.0013$ | $0.02861 \pm 0.00139$ | $0.08375 \pm 0.00187$ | $0.0777 \pm 0.00338$ |
| | Ours ($K=100$) | $\mathcal{O}(tK)$ | $0.04288 \pm 0.0012$ | $0.02783 \pm 0.00108$ | $0.08468 \pm 0.00172$ | $0.07658 \pm 0.00195$ |
| | Ours ($K=N$) | $\mathcal{O}(tK)$ | $0.0468 \pm 0.00086$ | $0.02149 \pm 0.00202$ | $0.09482 \pm 0.00117$ | $0.06081 \pm 0.00676$ |
| Steam | SASRec | $\mathcal{O}(N)$ | $0.14763 \pm 0.00051$ | $0.0 \pm 0.0$ | $0.18259 \pm 0.00055$ | $0.0 \pm 0.0$ |
| | FDSA | $\mathcal{O}(N)$ | $0.08236 \pm 0.00152$ | $0.0 \pm 0.0$ | $0.14773 \pm 0.00234$ | $0.0 \pm 0.0$ |
| | S$^3$-Rec | $\mathcal{O}(N)$ | $0.14437 \pm 0.00127$ | $0.0 \pm 0.0$ | $0.18025 \pm 0.00222$ | $0.0 \pm 0.0$ |
| | UniSRec | $\mathcal{O}(N)$ | - | - | - | - |
| | RecFormer | $\mathcal{O}(N)$ | $0.14034 \pm 0.00123$ | $0.00120 \pm 0.00037$ | $0.17042 \pm 0.00275$ | $0.00319 \pm 0.0011$ |
| | TIGER | $\mathcal{O}(tK)$ | $0.15034 \pm 0.00064$ | $0.0 \pm 0.0$ | $0.18980 \pm 0.00135$ | $0.0 \pm 0.0$ |
| | Ours ($K=20$) | $\mathcal{O}(tK)$ | $0.14951 \pm 0.00158$ | $0.00512 \pm 0.00047$ | $0.19049 \pm 0.00234$ | $0.01466 \pm 0.0011$ |
| | Ours ($K=40$) | $\mathcal{O}(tK)$ | $0.15138 \pm 0.0011$ | $0.00298 \pm 0.00068$ | $0.19302 \pm 0.0018$ | $0.00829 \pm 0.0011$ |
| | Ours ($K=60$) | $\mathcal{O}(tK)$ | $0.15236 \pm 0.00074$ | $0.00258 \pm 0.00109$ | $0.19455 \pm 0.00154$ | $0.00701 \pm 0.00292$ |
| | Ours ($K=80$) | $\mathcal{O}(tK)$ | $0.15284 \pm 0.00059$ | $0.00226 \pm 0.00105$ | $0.19522 \pm 0.00143$ | $0.00637 \pm 0.00292$ |
| | Ours ($K=100$) | $\mathcal{O}(tK)$ | $0.15318 \pm 0.00049$ | $0.00222 \pm 0.00109$ | $0.19566 \pm 0.00132$ | $0.00637 \pm 0.00292$ |
| | Ours ($K=N$) | $\mathcal{O}(tK)$ | $0.15431 \pm 6e\text{-}05$ | $0.00175 \pm 0.00082$ | $0.19736 \pm 0.00086$ | $0.0051 \pm 0.00221$ |

Figure 7 presents the ablation results in terms of Recall@10 across four datasets: Beauty, Sports, Toys, and Steam. The performance is analyzed with respect to the number of candidates retrieved by the SID head (K).

First, comparing *TIGER* to *TIGER(T)*, we observe that *TIGER(T)* consistently performs the same or slightly better, demonstrating the positive impact of incorporating the item's text representation as input. However, the improvement is modest, indicating that while text representation is helpful, its contribution alone does not significantly enhance the model's performance.

Second, when comparing *Dense* retrieval with *Dense (SID)*, the results are similar, suggesting that the primary limitation of the dense retrieval method is not due to the type of representation used (ID vs. SID). This highlights that the bottleneck contributing to the performance difference lies elsewhere, possibly in the learning of SID.

LIGER, which combines *TIGER* and *dense* retrieval paradigms, exhibits a smooth interpolation between the performance of *TIGER* and *Dense*. This suggests that LIGER effectively leverages the strengths of both approaches to achieve robust performance across datasets. Notably, when the gra-

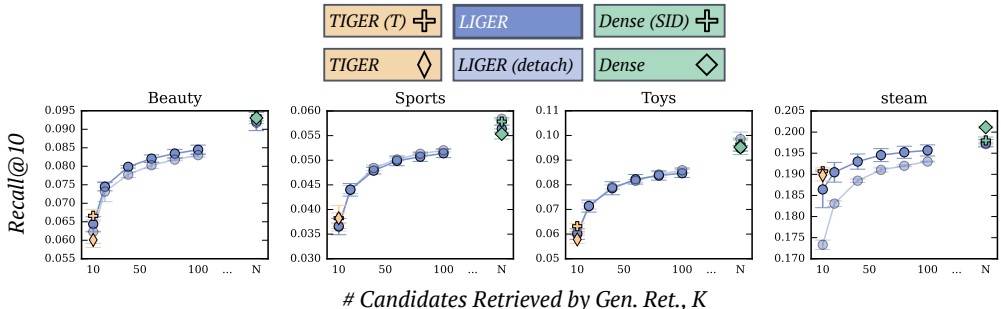

Figure 6: *Overview of our Ablation Study.* This study examines the effects of different components within LIGER (*top middle*), which integrates TIGER and semantic ID (SID)-based dense retrieval in an transductive setting. LIGER takes both the semantic ID and item text representation as inputs, predicting the SID and generating embeddings. We perform the following ablations to evaluate the impact of specific components: (1) To assess the effect of multi-objective optimization, we detach the gradient updates from the SID head (*bottom middle*). (2) To study the role of the embedding head, we remove it (*top left*). (3) To evaluate the contribution of the item text representation input in (2), we remove it, reducing the model to TIGER (*bottom left*). (4) To analyze the effect of the SID head, we remove it (*top right*). (5) Finally, we replace the SID with item IDs in (4), reducing the model to standard dense retrieval in transductive setting (*bottom right*).

Figure 7: *Ablation Results on Recall@10 across Datasets.*

dient update from the SID head is detached (LIGER*(detach)*), the model still performs comparably to the standard LIGER, with the most significant drop observed on the Steam dataset. This result implies that the SID head's learning signal is crucial for the Steam dataset, which is of a larger scale compared to the Amazon datasets.

Overall, these results demonstrate that LIGER strikes a balance between TIGER and dense retrieval methods while retaining flexibility through multi-objective optimization. However, the importance of the SID head's gradient signals appears dataset-dependent, possibly also influenced by dataset scales, as highlighted by the performance gap on Steam.

