# OpenReview forum: "Unifying Generative and Dense Retrieval for Sequential Recommendation"
_ICLR.cc/2025/Conference — Submitted to ICLR 2025_

### Official Review · Reviewer_s2kf · 2024-10-18

**Soundness:** 2
**Presentation:** 3
**Contribution:** 2
**Rating:** 5
**Confidence:** 3

**Summary:**

The paper introduces LIGER, a hybrid model that integrates generative and dense retrieval methods to improve recommendation systems. It presents a comparative study highlighting performance, storage, and computational trade-offs between the two methods, demonstrating significant enhancements, especially in handling cold-start items. The effectiveness of LIGER is validated through extensive experiments across multiple datasets, showing its superiority in performance and efficiency for large-scale applications.

**Strengths:**

1. Detailed Analysis and Comprehensive Approach: The paper provides a thorough analysis of the strengths and weaknesses of both generative and dense retrieval approaches. It introduces a comprehensive hybrid model, LIGER, that effectively combines the benefits of each method. This model addresses key challenges such as the cold-start problem and efficiency issues, demonstrating a strategic integration that enhances overall recommendation system performance.

2. Robust Experimental Validation: The study employs a robust set of datasets and baseline comparisons that thoroughly demonstrate the effectiveness of the proposed method. The detailed discussions on the number of candidates retrieved and the performance on cold-start items particularly validate the model's efficiency and effectiveness in managing these critical aspects. This comprehensive testing setup confirms the model's capability to maintain efficiency while improving cold-start performance, establishing a strong case for its practical application in large-scale environments.

**Weaknesses:**

1. Lack of Implementation Details and Code Availability: The paper does not provide specific formulas or code, making it difficult to replicate the study and understand how semantic IDs augment item representations. This lack of detailed methodology can hinder the ability of other researchers to verify results or extend the work.

2. Moderate Innovativeness: While the paper presents a novel hybrid model, the overall innovativeness may be viewed as moderate. The core idea of combining generative and dense retrieval is an extension of existing techniques rather than a groundbreaking discovery, which might limit the impact or interest among researchers looking for cutting-edge advancements.

**Questions:**

1. Could you provide more explicit formulas or algorithms that detail how semantic IDs augment item representations within the LIGER model?

2. Could you provide the analysis on how changes in key parameters such as the number of semantic IDs, beam search size, and the dimensions of embeddings affect the recommendation quality and computational efficiency.

---

### Official Review · Reviewer_BBri · 2024-10-24

**Soundness:** 2
**Presentation:** 1
**Contribution:** 2
**Rating:** 3
**Confidence:** 4

**Summary:**

This paper points out two questions in generative retrieval: performance gap and cold start problem. After that, this paper presents LIGER, which combines both generative and dense retrieval for sequential recommendation and addresses the cold start problem. Generally, the research topic of this paper is interesting, but the method of this paper fails to address the research problems. Moreover, the writing of this paper can be improved.

**Strengths:**

1. How to combine generative and dense retrieval is an important question.
2. The main experiment is comprehensive.

**Weaknesses:**

1. The motivation is unclear. The main claim in this paper is that (1) Generative retrieval performs worse than dense retrieval under the same amount of information. (2) Generative retrieval can hardly handle cold items. However, the verification experiment details are unclear (refer to weakness 2), making this claim unverified and somewhat overclaimed. Moreover, the cold start problem seems irrelevant to the first claim, making the structure of this paper divisive.

2. The method seems like a patchwork of generative and dense retrieval. More importantly, the proposed method does not seem to address the problems in both generative and dense retrieval. As illustrated in Table 1, LIGER needs to store O(t+N) embeddings (both learnable and fixed), which is higher than both generative and dense retrieval. Moreover, the inference cost is also huge, requiring a generative paradigm with beam search. Such a paradigm is actually much slower than the single dense retrieval [3]. Please also provide the inference time (in seconds) in the experiments.

    As a result, the proposed method is likely to combine the drawbacks of both generative and dense retrieval rather than optimizing them.

3. The paper writing can be significantly improved.

    2.1. The terminologies in this paper are not introduced clearly. (2.2.1), “learnable text representation” is introduced only in lines 243 and 244. I have searched this terminology globally but find no explanations for it. Moreover, it is unclear whether “learnable text representation” is the same as “learnable embedding”. (2.2.2), “the same amount of information” seems to be an important condition for evaluating the performance of generative and dense retrieval, but there are also no explanations for it. (2.2.3)

    2.2. There are not any formulations in this paper, including both task formulation and method formulation. Such a lack of formulation makes this paper hard to read.

    2.3. The structure of this paper is unnatural. To verify the two significant claims in this paper, please detail on the implementation details of the verification experiments first, rather than putting all the experiment settings in the experiment section. Please refer to previous papers [1,2] to learn how to write such findings-and-method papers.

    2.4. The descriptions of figures in this paper are wordy and confusing. Please refer to weakness 2.3 to formulate the method and task in math and use precise and condensed sentences to describe the figure. Moreover, please detail the method in the main body of this paper with mathematical formulation to make your method more readable.

    The chaotic paper writing makes it hard to understand the details of this paper. I suggest the authors improving the writing of this paper to make it more clear.


[1] Liang, Victor Weixin, et al. "Mind the gap: Understanding the modality gap in multi-modal contrastive representation learning." *Advances in Neural Information Processing Systems* 35 (2022): 17612-17625.

[2] He, Xiangnan, et al. "Lightgcn: Simplifying and powering graph convolution network for recommendation." *Proceedings of the 43rd International ACM SIGIR conference on research and development in Information Retrieval*. 2020.

[3] Rajput, Shashank, et al. "Recommender systems with generative retrieval." *Advances in Neural Information Processing Systems* 36 (2024).

**Questions:**

Refer to the weaknesses. My main concerns focus on the motivation of this work, which is not clearly presented and addressed.

---

### Official Review · Reviewer_BZ1a · 2024-10-28

**Soundness:** 2
**Presentation:** 3
**Contribution:** 2
**Rating:** 5
**Confidence:** 5

**Summary:**

This paper presents a comparison between generative retrieval and sequential dense retrieval models for sequential recommendation. In the paper, the authors identify two key limitations of generative retrieval: a performance gap compared to dense retrieval, and a tendency to overfit to training items, resulting in lower probability of generating cold-start items. In order to address these challenges, the authors propose LIGER, a hybrid model that combines the strengths of dense and generative retrieval, which is built upon TIGER. LIGER leverages the computational and storage efficiencies of generative retrieval, while enhancing its capabilities in generating cold-start items and improving ranking performance through the integration of dense retrieval techniques. Extensive experiments show that LIGER outperforms existing sequential dense and generative retrieval models on benchmark datasets, and demonstrates better performance in scenarios involving cold-start items.

**Strengths:**

1. The paper presents a comprehensive comparison between generative retrieval and sequential dense retrieval models for sequential recommendation, identifying key limitations of generative retrieval, with clear illustrations and tabulates the computational cost.
2. The authors propose LIGER, a hybrid model that combines the strengths of dense and generative retrieval to address the limitations of generative retrieval, reducing the performance gap and improving the generation of cold-start items.
3. The authors have conducted a series of experiments comparing with most of the existing SOTA sequential retrieval methods, and show significant performance gain of their methods.

**Weaknesses:**

My main concern of the papers are as follows:
1. Novelty. Despite performance improvement, the paper is a simple extension of the TIGER method which the combination of dense retrieval, which is quite straightforward and hence the contribution in terms of the methodology is largely incremental.
2. Dataset. The choice of the datasets are not optimal. All four datasets are pretty old and relatively small, comparing to recent released datasets for recommendation tasks, such as KuaiRand, KuaiRec and KuaiSAR etc (https://kuairand.com/). The biggest dataset used in the paper only contains 599,620 actions and 400 cold-start items, letting alone other three datasets. With the models getting more and more sophisticated and larger, it is quite easy to overfit on these small(tiny) datasets. Hence I'm not convinced by the experiments.
3. Ablation study. LIGER consists of several modules. It is necessary to show the performance gap when each individual module is removed.

**Questions:**

Please clarify the weaknesses that I listed above, so we can discuss about potential improvement of the paper.

---

### Official Review · Reviewer_Emoa · 2024-11-01

**Soundness:** 2
**Presentation:** 2
**Contribution:** 2
**Rating:** 5
**Confidence:** 3

**Summary:**

This paper proposes a new hybrid retrieval model LIGER (LeverageIng dense retrieval for GEnerative Retrieval) to improve the performance of sequential recommendation systems. Traditional dense retrieval methods rely on dense embeddings for recommendations. Although they are highly accurate, they have high computational and storage costs. Generative retrieval models directly predict item indexes through generative models and use semantic IDs to improve storage efficiency, but they have certain limitations in performance and cold start recommendations. The LIGER model combines the candidate generation capabilities of generative retrieval and the precise ranking of dense retrieval, aiming to balance recommendation performance and efficiency, especially to deal with the cold start problem.

**Strengths:**

Rigorous research process: The paper conducts detailed experimental verification for each core hypothesis and key proposition. Through the corresponding experimental data, the advantages of the LIGER model in recommendation accuracy, cold start processing ability, and the quality of generated candidate sets are demonstrated.

Research motivation : The research motivation of the paper is clear, and it studies two key issues in the recommendation system: how to strike a balance between performance and efficiency, and how to deal with the cold start problem.

**Weaknesses:**

The chart design is not intuitive enough: The design of the chart (especially the Normalized Performance Gap related chart) has room for improvement in information presentation. The captions are too long and the content is dense, resulting in unclear information transmission and affecting readability. It is recommended to optimize the structure and presentation of the chart to more intuitively show the impact of different numbers of candidate items on the performance gap between generative retrieval and intensive retrieval.

Lack of formula support and insufficient rigor in method expression: The core method part of the paper lacks the support of mathematical formulas and mainly relies on text descriptions and diagrams, which may cause readers to have difficulties in understanding the specific implementation details of the model. It is recommended to introduce formulas in the method part to systematically show the key calculation steps and algorithm structure of the LIGER model, so as to enhance the accuracy and clarity of expression.

Lack of experiments: Although the paper analyzes the theoretical complexity difference between generative retrieval and dense retrieval, it lacks intuitive data on actual running time. For example, the running time under specific datasets and different K values ​​(number of candidates) is not shown. In addition, the choice of K value directly affects the balance between efficiency and performance of the model, but the sensitivity of this hyperparameter is not explored in the paper. It is recommended to supplement the running time measurement of different K values ​​on specific datasets and analyze the specific impact of K value on performance and efficiency, so as to provide a reference for more reasonable selection of K value in practice.

**Questions:**

In the paper, Normalized Performance Gap (NPG) is used to measure the performance gap between Generative Retrieval and Dense Retrieval. However, the definition and actual performance of NPG have caused some confusion, especially in the interpretation of the changing trend when the number of candidates $K$ increases.Specifically, as the number of candidates $K$ increases, the NPG value in the figure rises and stabilizes, which seems to indicate that the gap between generative retrieval and dense retrieval is widening. Does this trend mean that the performance gap between generative retrieval and dense retrieval widens with the increase of $K$, or does it reflect that generative retrieval is gradually approaching the performance of dense retrieval? If it is the latter, it is recommended to further clarify the definition of NPG and explain the exact meaning of the increase in NPG.

In addition, if the definition of NPG is based on the normalized gap between generative retrieval and dense retrieval in indicators such as Recall@10, please consider clearly giving the calculation formula in the method section so that readers can better understand the actual meaning of NPG.

---

### Official Review · Reviewer_Yepi · 2024-11-02

**Soundness:** 2
**Presentation:** 2
**Contribution:** 2
**Rating:** 3
**Confidence:** 5

**Summary:**

This paper proposes a method that combines generative and discriminative recommendations to improve the performance of generative recommendation systems and address the cold start problem. Specifically, the authors integrate TIGER and Recformer to some extent and demonstrate good results on three datasets.

**Strengths:**

1. The idea of combining generative and discriminative approaches is promising.
2. The method presented by the authors is relatively simple and easy to understand and follow.
4. Experiments are conducted on four datasets, showing good performance.

**Weaknesses:**

1. There are issues with some claims or descriptions in the introduction. For instance, generative recommendation methods are not limited to codebook-based approaches like TIGER, yet the authors present it as the only form of generative retrieval; also, generative methods are not inherently incapable of handling cold start recommendations since some works can solve this problem. Additionally, the authors do not introduce the core idea of their method in the introduction. And few studies in the recommendation domain use the terms "generative retrieval" and "sequential dense retrieval".
2. The authors seem to partially integrate TIGER and Recformer; however, why not simply ensemble the two models to combine their advantages? This approach should at least be compared in experiments.
3. The experimental settings are unclear, particularly regarding the cold start evaluation. It is not specified whether negative sampling or full ranking was used. Given that TIGER also utilizes semantic encoding, it should not yield entirely zero performance.
4. The number of cold start items in the datasets is low, raising questions about statistical significance.
5. The proposed LIGER model does not appear to be very robust on different datasets.
6. The authors should compare their method with more recent works beyond TIGER and Recformer, as there are now many updated approaches in this area.
7. Comparing the computational costs of models like TIGER and Recformer requires taking into account that generative methods must generate over the entire vocabulary. Each step's computational load depends on the vocabulary size, and the generation is performed autoregressively, step by step, which may not result in high time efficiency in experiments.

**Questions:**

1. Why did the authors choose to integrate TIGER and Recformer rather than using an ensemble approach? Would an ensemble be more effective in leveraging the strengths of both?
2. Could the authors clarify the cold start evaluation settings?
3. How significant are the results given the limited number of cold start items in the datasets?

---

### Meta-Review · Area_Chair_6Lwm · 2024-12-22

**Metareview:**

This paper studies the sequential recommendation problem. The authors propose to combine the generative and discriminative recommendation methods to improve the performance of the generative recommendation systems. Speciffically, they propose to combine TIGER and Recformer to achieve good recommendation results on 4 public datasets.

For this paper, the technical contribution of the proposed method seems limited, and the motivation is not very clear. There also exist some overclaims in this paper. It seems that the proposed does not well address the challenges of both the generative and discriminative recommendation methods. Moreover, the experimental datasets are a little small. The authors need to perform experiments on datasets that include millions of user-item interactions. The number of cold-items in the experiment datasets are 43, 56, 81, and 400. This is not sufficient to demonstrate the effectiveness of the proposed method in handling cold-start problem. Moreover, all the experiments only include the user explicit ratings. The authors are suggested to include some datasets that include users' implicit feedback (e.g., clicking) for experimental evaluation, e.g., Kuaishou dataset.

**Additional Comments On Reviewer Discussion:**

In the rebuttal, the authors and reviewers have discussed about the motivation of the proposed method and some overclaims in the papers. The authors have made some modifications based on the user comments. Moreover, the authors also provide additional experiments on Amazon Musica Instrutments to demonstrate the robustness of the proposed method. They also provide some explanation about the NPG definition, which helps the reviewer get a better understanding of this work.

The authors claimed they perform experiments on the Steam dataset, which includes 2,567,538 users, 15,474 items, and 7,793,069 interactions. However, only 47,761 users, 12,012 items, and 599,620 interactions are used in the experiments. For the largest experiment dataset, it only includes about 600 thousands of user-item interactions. The authors need perform some experiments on large-scale datasets, which include millions of user-item interactions.

---

### Decision · Program_Chairs · 2025-01-22

Reject